A biogeographic framework of octopod species diversification: the role of the Isthmus of Panama

Lima Francoise D. francoisedl@yahoo.com.br 1
Strugnell Jan M. 2
Leite Tatiana S. 3
Lima Sergio M.Q. 1
1 Department of Botany and Zoology, Universidade Federal do Rio Grande do Norte , Natal , Rio Grande do Norte , Brazil
2 Centre for Sustainable Tropical Fisheries and Aquaculture, James Cook University , Townsville , Queensland , Australia
3 Department of Ecology and Zoology, Universidade Federal de Santa Catarina , Florianópolis , Santa Catarina , Brazil
Bieler Rudiger
Electronic publication date: 2020 Mar 27
Publication date: 2020
Volume: 8
Electronic Location ID: e8691
Received 2019 Jun 14; Accepted 2020 Feb 5
Copyright: ©2020 Lima et al.
Copyright year: 2020
Copyright holder: Lima et al.
License: This is an open access article distributed under the terms of the Creative Commons Attribution License, which permits unrestricted use, distribution, reproduction and adaptation in any medium and for any purpose provided that it is properly attributed. For attribution, the original author(s), title, publication source (PeerJ) and either DOI or URL of the article must be cited.
License URL: https://creativecommons.org/licenses/by/4.0/

Keywords: Phylogeny, Vicariance, Dispersal, Speciation, Fossil calibration, Octopus, Isthmus of Panama, Evolution

Funding: Brazilian National Research Council CNPq 481492/2013-9 Coordination for the Improvement of Higher Education Personnel CAPES, Projeto Ciências do Mar II- 23038.004807/2014-01 Françoise Dantas Lima to develop the work in Brazil and in Australia through the program Science Without Borders James Cook University This work was supported by the Brazilian National Research Council (CNPq 481492/2013-9) and Coordination for the Improvement of Higher Education Personnel (CAPES, Projeto Ciências do Mar II- 23038.004807/2014-01). They provided a grant for the Françoise Dantas Lima to develop the work in Brazil and in Australia through the program Science Without Borders. This work also received support from James Cook University, mediated by Dr. Jan Strugnell. The funders had no role in study design, data collection and analysis, decision to publish, or preparation of the manuscript.

==============================
The uplift of the Isthmus of Panama (IP) created a land bridge between Central and South America and caused the separation of the Western Atlantic and Eastern Pacific oceans, resulting in profound changes in the environmental and oceanographic conditions. To evaluate how these changes have influenced speciation processes in octopods, fragments of two mitochondrial (Cytochrome oxidase subunit I, COI and 16S rDNA) and two nuclear (Rhodopsin and Elongation Factor-1α, EF-1α) genes were amplified from samples from the Atlantic and Pacific oceans. One biogeographical and four fossil calibration priors were used within a relaxed Bayesian phylogenetic analysis framework to estimate divergence times among cladogenic events. Reconstruction of the ancestral states in phylogenies was used to infer historical biogeography of the lineages and species dispersal routes. The results revealed three well-supported clades of transisthmian octopus sister species pair/complex (TSSP/TSSC) and two additional clades showing a low probability of species diversification, having been influenced by the IP. Divergence times estimated in the present study revealed that octopod TSSP/TSSC from the Atlantic and Pacific diverged between the Middle Miocene and Early Pliocene (mean range = 5–18 Ma). Given that oceanographic changes caused by the uplift of the IP were so strong as to affect the global climate, we suggest that octopod TSSP/TSSC diverged because of these physical and environmental barriers, even before the complete uplift of the IP 3 Ma, proposed by the Late Pliocene model. The results obtained in this phylogenetic reconstruction also indicate that the octopus species pairs in each ocean share a recent common ancestor from the Pacific Ocean.

Introduction

The formation of the Isthmus of Panama (IP) caused profound changes in environmental and oceanographic conditions (Haug & Tiedemann, 1998; Bartoli et al., 2005; Schneider & Schmittner, 2006), which influenced dispersal and speciation processes in terrestrial and marine biota (Lessios, 1981; Knowlton & Weigt, 1998; Leigh, O’Dea & Vermeij, 2013). The closure of the connection between the Atlantic and Pacific oceans is considered the most important vicariant event of the Cenozoic (O’Dea et al., 2016), providing a remarkable system to study evolutionary processes in a natural environment.

The age of the final closure of the IP, as well as its role in fundamental evolutionary processes is controversial (Stone, 2014), and two hypotheses were proposed to describe this event: the Late Pliocene and the Middle Miocene models. Using geological, fossil and molecular data, several authors have proposed that the final closure of the isthmus was in the Late Pliocene, approximately 2.5–3.5 millions years ago (Ma) (Coates et al., 1992; Bartoli et al., 2005; Schneider & Schmittner, 2006; Coates & Stallard, 2013; O’Dea et al., 2016). However, recent studies based on dispersal waves of terrestrial organisms and geochronological information have suggested that the closure of the this seaway occurred during the Miocene (15 Ma) (Bacon et al., 2015; Hoorn & Flantua, 2015; Montes et al., 2015).

According to the Late Pliocene hypothesis, this long process initiated with a collision of Central and South America about 15–24 Ma ago and the formation of a volcanic arc, around the Early Miocene (Coates & Stallard, 2013). During the Middle Miocene (around 10 Ma), successive collisions caused widespread shallowing of the oceans and major changes of oceanic conditions, and deep and intermediate water exchanges between the Atlantic and Pacific were closed (Coates et al., 2004; Keigwin, 1982). Around 6–4 Ma oceanic conditions, including temperature, salinity, sedimentary carbon content and habitat availability on each side of the isthmus changed substantially (Haug & Tiedemann, 1998; Leigh, O’Dea & Vermeij, 2013). By 3 Mya, the uplift of the IP completely separated the waters of the Tropical Western Atlantic and the Tropical Eastern Pacific (Coates & Stallard, 2013; O’Dea et al., 2016).

Studies based on Uranium-lead geochronology in detrital zircons in the Andes, Panamanian fluvial deposits and inferences on terrestrial and aquatic dispersal provide a different insight for the early closure of the IP; the Middle Miocene hypothesis (Montes et al., 2012; Bacon et al., 2013; Bacon et al., 2015; Montes et al., 2015). In this approach, the isthmus formation started around 38–28 Ma, and the collision between the southern tip of Central America and South America occurred during the Late Oligocene (28.1–23.0 Ma). Montes et al. (2015) suggested that the complete closure interrupting the water connection between the Eastern Pacific and Western Atlantic occurred around 14–15 Ma.

The timing of the uplift of the IP, based mainly on the Late Pliocene hypothesis, is widely used as a biogeographical calibration point and is considered one of the most important geological events for calibrating molecular clocks (Lessios, 2008; Parham et al., 2012; Gleadall, 2013). However, to evaluate the influence of the final seaway closure on divergence and distribution of Atlantic and Pacific sister species, it is important to use a calibration independent of the isthmus formation to avoid circular reasoning in the divergence time interpretation (Marko, 2002; O’Dea et al., 2016). Therefore, molecular phylogenies containing sister species from each side of the IP have been calibrated using the fossils record and/or the molecular evolutionary rate of a particular gene (e.g., Bermingham & Lessios, 1993; Knowlton et al., 1993; Collins, 1996; Bartoli et al., 2005; Bacon et al., 2013; Galván-Quesada et al., 2016).

The emergence of geographical barriers, such as the formation of the Isthmus of Panama, may reduce or interrupt gene flow between populations. The vicariant populations can evolve different genotypic characteristics over time, although they can retain similar morphological and behavioural characteristics (Palumbi, 1994; Knowlton, 2000; Cowman & Bellwood, 2013). The species that diverged as a consequence of the uplift of the Isthmus of Panama are closely related morphologically and genetically and are called transisthmian sister species pairs/complex (TSSP/TSSC) (Lessios, 2008; Marek, 2015; Marko, 2002). According to the transisthmian sister species complex concept, several species on one side of the Isthmus can represent the putative sister group of the species on the other side (Marek, 2015).

Transisthmian clades including isopods, echinoids, crustaceans, fishes and molluscs have already been identified on each side of the Isthmus of Panama based on molecular and fossil calibrated phylogenies (Graham, 1971; Lessios & Weinberg, 1994; Lessios, 2008; Marko, 2002; O’Dea et al., 2016). Nesis (2003) stated that there are no cephalopod species occurring on both sides of the IP, although nine pairs of similar species are known in the Eastern Pacific and Western Atlantic: two pairs of shallow-water squids (Family: Loliginidae) and seven pairs of benthic octopuses (Family: Octopodidae). According to Voight (1988), the high degree of similarity among pairs of pygmy, ocellated, common and striped octopod species, distributed along either side of Central America, is evidence that each pair shares a common ancestor, which is a more parsimonious explanation than convergent evolution.

The Superfamily Octopodoidea includes the families Octopodidae, Bathypolypodidae, Enteroctopodidae and Megaledonidae, which encompass a high diversity of benthic octopus (Boyle & Rodhouse, 2005; Norman, Hochberg & Finn, 2014; Sanchez et al., 2018). Some species have a planktonic phase before settling on the substrate and are characterised by high fecundity (20,000–500,000 eggs) and small eggs (20–50 mm). Other species have low fecundity (50–800 eggs), large eggs (6–18 mm) and hatchlings that settle directly on the seabed (Mangold, 1987; Iglesias et al., 2007). The oceanographic characteristics such as temperature, ocean currents and productivity are important determinants of trait evolution of larvae and their dispersal ability (Robertson & Collin, 2015). For this reason, the contrasting life history traits among octopod species make them an interesting model to study the processes of speciation and adaptive divergence after environmental changes caused by a vicariant event.

The consequences of forming the Isthmus of Panama were dramatic to marine biodiversity, ocean circulation and global climate. Thus, this study aims to understand how the uplift of the IP and the environmental changes caused by this vicariant event influenced speciation and dispersal processes of octopod species in the Atlantic and Pacific oceans. Based on that, Bayesian time-calibrated (biogeographical and fossil) phylogenetic analysis was performed to identify putative transisthmian sister species pairs/complexes and verify whether the divergence time among the octopod lineages coincides with the Pliocene or Miocene hypotheses.

Material & Methods

Data collection

Tissue samples of octopod species were collected by snorkelling and SCUBA diving in Brazil (North-eastern coast and four oceanic islands) (permit SISBIO 10706-5 and 30484-1), from fish markets and landings in Mexico (Isla Mujeres, Sisal and port of Progreso) (Fig. 1) (Table S1). Muscle tissue samples were taken from octopod arms preserved in 95% ethanol and stored at −20°C.

Figure 1 Localities of the specimens used in this study to estimate phylogenetic relationships and divergence times among octopods species.

Orange circles represent species from the Eastern Pacific (EP), blue circles from the Western Atlantic (WA), green circles from the Eastern Atlantic (EA), and pink circles are species from the Western Pacific (WP). The Isthmus of Panama is indicated with a red square.

Initial analyses were performed using sequences generated in the present study, and additional sequences were obtained from GenBank in order to evaluate the closest phylogenetic relationships among putative octopods sister species occurring in the Atlantic and Pacific. A total of 135 sequences from 30 cephalopods species (70 from this study and 65 obtained from GenBank) were chosen to estimate divergence times and infer phylogenetic relationships. The sequences of genes fragments generated in this study are accessible from GenBank under accession numbers (COI: MN933632 –MN933651; 16S: MN508063 –MN508082; rhodopsin: MN946381 –MN946396; EF-1α: MN946371, MN946372, MN946373, MN946374, MN946375, MN946376, MN946377, MN946378, MN946379, MN946380) (Table S1).

Calibration priors

Four fossils and also biogeographical information (Table 1) were used in order to calibrate the species tree. Calibration priors used include:

Table 1 Details of fossil and biogeographical information used to calibrate the phylogeny.

The prior probability distributions and posterior probability densities after the Monte Carlo Markov Chain (MCMC) run are showed.

Calibration node	Distribution	Type	Prior (Ma)	MCMC results (Ma)	
			Mean	Offset	95% CI	Mean	95% HPD	
M. longibrachus akambei×M. longibrachus longibrachus	Normal	Biogeographical	0.025	–	0.015, 0.035	0.032	0.024, 0.04	
Argonauta nodosa×Tremoctopus violaceus	Exponential	Fossil	35	29	30, 62	46	29, 68	
Cirrata × Incirrata	Exponential	Fossil	25	90	95, 187	101	90, 121	
Vampyromorpha × Octopoda	Exponential	Fossil	24	162	162, 250	170	162, 187	
Notes.

Ma millions of years ago

CI confidence interval

HPD Highest Posterior Density

1—The biogeographic prior was the separation of deep sea Muusoctopus longibrachus subspecies (Muusoctopus longibrachus longibrachus Ibáñez, Sepúlveda & Chong, 2006 and Muusoctopus longibrachus akambei Gleadall et al., 2010) as a result of events related to the Last Glacial Maximum (LGM) proposed by Gleadall (2013). A normal prior was set on the M. longibrachus node with a mean of 24 ± 5 kya, yielding a range between 5% and 95% quartiles of 15–33 kya (Gleadall, 2013).

2—Divergence between Argonauta and Tremoctopus. An exponential prior was used based on the earliest record of the Argonautidae, the fossil Obinautilus pulcher from the Oligocene (29 Ma) (Kobayashi, 1954). The upper bound of 64 Ma was placed at the last occurrence of ammonites as a prior 95% confidence interval (CI) of the distribution of Argonauta. In the Cretaceous, ammonite shells were abundant but were not recorded in the Tertiary. According to Strugnell et al. (2008), although ammonites and argonauts have similar shells, there are no Argonauta fossil records as old as ammonite records. Therefore, it is conservative to set this upper bound on the prior 95% CI of the distribution of argonauts at the last occurrence of ammonites, 36 million years prior to their appearance in the fossil record.

3—Split between the suborders Cirrata (Opisthoteuthis massyae) and Incirrata based on the fossils of Keuppia levante, K. hyperbolaris and Styletoctopus annae from the Upper Cenomanian, between the Toarcian (180 Ma) and the early Turonian (95 Ma). These fossils are regarded as the earliest representatives of the Incirrata (Fuchs, Bracchi & Weis, 2009). This is represented by an exponential prior on the MRCA of these groups, with the minimum age of 95 Ma and 180 Ma as an upper bound on the prior 95% confidence interval.

4—Separation of Vampyromorpha (Vampyroteuthis infernalis) and Octopoda. An exponential prior with lower bound of 162 Ma was chosen based on the fossil Vampyronassa rhodanica from the Lower Callovian of the Jurassic (Fisher & Riou, 2002). The upper bound of 250 Ma was based on studies of Strugnell et al. (2006) and Kröger, Vinther & Fuchs (2011), who affirm that Vampyropods diverged by or before the Permian.

Phylogenetic analysis

Genomic DNA was extracted using the GF-1 Nucleic Acid Extraction kit (Vivantis, Malaysia) according to the manufacturer’s instructions. Fragments of two mitochondrial (Cytochrome oxidase subunit I, COI and 16S rDNA) and two nuclear (Rhodopsin and Elongation Factor-1α, EF-1α) genes were amplified in this study. Cytochrome oxidase I gene amplicons were obtained using universal primers LCO1490 and HCO2198 (Folmer et al., 1994), and partial sequences of 16S rDNA were amplified with the primers 16SarL and 16SbrH (Palumbi, 1996). The forward and reverse primers for amplification of Rhodopsin (RhFwd1 5′ GATCGTTACAATGTCATCGGTAGACC 3′, RhRev4 5′ GAGAAAGAATGCGAAGATGCTA 3′) and EF-1α (EFFwd1 5′ TCTGGTTGGCATGGTGATAACATG 3′, EFRev3 5′ ATTGTCATTAACCACCCTGGAC 3′) were designed from octopus sequences available on GenBank using the software Geneious 9.0.2 (Kearse et al., 2012).

The PCR amplification reactions of all sequences were conducted in a final volume of 25 µL containing 1 µL forward primer, 1 µL reverse primer (10 mM), 12.5 µL Taq DNA Polymerase Master Mix (Ampliqon A/S) or MyTaq RedMix (Bioline), 8.5 µL H2O and 2 µL DNA (20–40 ng/µl). PCR cycle parameters used to amplify COI and 16S genes were 3 min at 95 °C for denaturation, followed by 35 cycles of 1 min at 94 °C, 1 min at 45 °C for annealing, 1.5 min at 72 °C for extension and a final extension step of 4 min at 72 °C. The parameters used to amplify Rhodopsin, detailed in Allcock, Strugnell & Johnson (2008), were the same used to amplify EF-1α. The PCR products were purified and sequenced by Macrogen Inc, Seoul, Korea.

Electropherograms were edited with Geneious 9.0.2 (Kearse et al., 2012), and sequences were aligned by ClustalW using Mega 6 (Tamura et al., 2013). Unalignable loop regions of 16S rDNA and gaps of EF1-alpha were removed before analysis using Gblocks software (Castresana, 2000). The substitution model for each gene was chosen on the basis of the hierarchical and Akaike information criterion tests using the software jModeltest (Posada, 2008). The substitution models most suitable for each gene according to both tests were GTR+G (COI), GTR+G+I (16S), and HKY85+G (rhodopsin and EF1-alpha).

Bayesian phylogenetic inference on the subset of sequences was carried out in BEAST 1.8.4 (Drummond et al., 2012). A total of 33 specimens (COI—33, 16S rDNA—32, rhodopsin—27, EF1-alpha—8) were included in subsequent analyses as separate partitions with unlinked substitution models and linked clock and tree models. An uncorrelated lognormal relaxed clock model incorporating Yule speciation-process prior on branching rates was used. Monte Carlo Markov Chain (MCMC) runs were performed for 3 × 108 generations, sampling one tree each 3 × 104 runs. The convergence of MCMC runs, effective sample size and the correct ‘burn-in’ for the analysis were assessed using Tracer v1.6 (Rambaut et al., 2014). A consensus tree accessing the posteriori probability values of each clade was generated using TreeAnnotator 1.8.3 (Drummond et al., 2012) and displayed using FigTree 1.4.3.

Biogeographical analysis

The distribution ranges of the species were divided into four areas: Western Atlantic (WA), Eastern Atlantic (EA), Western Pacific (WP) and Eastern Pacific (EP) (Fig. 1).

Reconstruct Ancestral State in Phylogenies (RASP) software package (Yu et al., 2015) was used to infer historical biogeography of the lineages, based on present-day distributions of cephalopods species (Norman, Hochberg & Finn, 2014). We used the Bayesian Binary Method (BBM) to estimate the probabilities of ancestral ranges by calculating the average probability of presence (1) and absence (0) over all sampled generations of the ancestral species in each area (Yu et al., 2015). Events of vicariance, dispersal and extinction, as well as the route of dispersal on the nodes, were also calculated by BBM model in RASP. The analysis was carried out with the Maximum Clade Credibility Tree (MCCT) from BEAST phylogenetic reconstruction, setting four heated MCMC chains, which run simultaneously for 5 × 106 generations, sampled every 1,000 generations.

The Time-Event Curve (TEC) was obtained by re-calculating the time of each node using the time of the root. The events on the node were treated using a modified Gaussian distribution (Yu et al., 2015). Thus, events of extinction, dispersal and vicariance are assigned to a time frame. This analysis was also carried out using RASP.

All the clades considered as transisthmian must contain species from each side of the IP and must have diverged after a vicariance event indicated by the RASP analysis.

Results

A total of 1,992 bp of the combined dataset (COI—609, 16S—396, rhodopsin—546, EF—441) were used to infer phylogenetic relationships and divergence times among 30 cephalopod species.

Phylogenetic analysis

The Bayesian phylogenetic tree, built using a relaxed phylogenetic approach, reveals three well-supported clades of transisthmian octopus species pair/complex (clades 1, 3, 5) that underwent vicariance processes as indicated by the RASP analysis (Fig. 2). Two other clades were also recovered, in which the divergence processes among the species showed a low probability of having been influenced by the IP (clades 2 and 4) (Fig. 2). The divergence time estimation indicated that the mean age of separation among TSSP/TSSC varied from 5.22 to 17.24 Ma (Table 2).

Figure 2 Integrated Bayesian phylogenetic tree and reconstruction of ancestral state.

The Bayesian posterior probabilities of the clades are shown below the nodes. Pie charts show the posterior probabilities of ancestral areas on different nodes. The five clades discussed in this study are highlighted in the phylogeny. The transisthmian sister species pair/complex are indicated by an asterisk in the clades 1, 3 and 5. Gray area represents the interval of divergence time found in octopod species (5.22–17.24). The circles surrounding some pie charts on the nodes indicate vicariance events. The graph shows events of dispersal and vicariance (axis y) assigned to a time frame as a result of a modified Gaussian distribution. Vampyroteuthis infernalis is not shown here due to its long branch length (see Fig. S1).

Clade 1 is composed by the deep sea Muusoctopus species (Family: Enteroctopodidae) and is characterised by two vicariant events. The first occurred 11.39 million years ago (5.08, 20.77 95% HPD) at the divergence of M. januari (WA) and a clade containing M. yaquinae (EP) and M. longibrachus subspecies (Posterior probability [PP] = 1). The second vicariance event corresponds to the divergence of M. longibrachus longibrachus and M. longibrachus akambei.

Clade 2 comprises the nocturnal species Callistoctopus ornatus from WP and Callistoctopus sp. and C. macropus from WA and EA, respectively (PP = 1). The divergence between those two lineages from Atlantic and Pacific was estimated to occur 5.22 Ma (2.21, 10.96 95% HPD).

Table 2 Divergence time estimates for each clade containing transisthmian sister species pair/complex.

The events of dispersal and vicariance are shown for each clade. The possible routes of dispersal for the most recent common ancestor in each clade are also shown with their respective probabilities.

Clade	Divergence time (Ma)	Vicariance event	Dispersal event (N)	BBM Route	BBM Probability	
	Mean	95% HPD					
1	11.39	5.08, 20.77	1	2	EP->WA	0.8801	
2	5.22	2.21, 10.96	1	2	WP->WA	0.3678	
3	17.24	8.99, 30.31	1	2	EP->WA	0.8745	
4	7.83	2.82, 15.88	1	2	WP->WA	0.8771	
5	8.03	4.27, 13.58	1	2	EP->WA	0.8739	
Notes.

EP Eastern Pacific

WP Western Pacific

WA Western Atlantic

Ma millions of years ago

BBM Bayesian Binary Method

HPD Highest Posterior Density

The pygmy octopuses (Paroctopus) species that are distributed in both sides of the Isthmus of Panama formed a highly-supported monophyletic clade (clade 3, PP = 1). These Paroctopus species diverged from the most recent common at 17.44 Ma (8.99, 30.31 95% HPD).

The long-armed mimic octopuses Macrotritopus defilippi from the Western Atlantic and Octopodidade sp. (White V, as referred by Norman, 2000) from the Western Pacific are sister taxa (PP = 1) and are estimated to have diverged 7.83 Ma (2.82, 15.88 95% HPD), as shown in clade 4.

The transisthmian sister species complex in clade 5 form a well-supported monophyletic group (PP = 1) and were estimated to have had a common ancestor 8.03 Ma (4.27, 13.58 95% HPD) (PP = 1). In this clade, O. insularis, O. maya and O. hummelincki from WA are the sister taxa to O. mimus and O. hubbsorum from EP and seem to have diverged after a vicariance event, indicated by the RASP analysis.

Biogeographical analysis

Reconstruct Ancestral State in Phylogenies (RASP) showed that the most recent common ancestors of the TSSP/TSSC originated in the Pacific Ocean (Table 2). Clades 1, 3 and 5 had high probabilities of an ancestral distribution in the East Pacific (PP = 0.88, 0.87, 0.87, respectively). Clades 2 and 4 were estimated to have originated from ancestors distributed in the Western Pacific Ocean (PP = 0.37 and 0.88, respectively). The dispersal routes for each TSSP/TSSC clade estimated by Bayesian Binary Method are shown in the Table 2.

The biogeographical analysis revealed five vicariant and ten dispersal events on the well-supported nodes of five transisthmian species pair/complex during the early Miocene to early Pliocene (Table 2). The time-event curve (TEC) analysis shows an increase of vicariant and dispersal events after the middle Miocene (∼15 Ma), with a peak in the Early Pliocene (∼5 Ma).

Discussion

Bayesian phylogenetic inference revealed transisthmian sister species pair/complexes in three genera of octopod species and low probabilities in two genera. These species diverged between 18 and 5 Ma, before the closure of the Isthmus of Panama suggested by the Late Pliocene model (3 Ma). The ancestral area reconstruction analysis showed vicariant events on each node of TSSP/TSSC and an increase of dispersal and vicariance during the process of isthmus formation, which indicates an influence of the emergence of the geological barrier (Isthmus of Panama) on the divergence processes among octopod species. In addition, the RASP results suggest that the most recent common ancestor of the five clades occupied the Pacific Ocean, and the most probable route of dispersal before the closure of the IP is from Pacific to Atlantic.

The divergence times estimated in the present study revealed that octopods TSSP/TSSC from Atlantic and Pacific diverged between the Middle Miocene and Early Pliocene (mean range = 5–18 Ma). This suggests that after 15 Ma (age of the isthmus final closure in the Miocene model), there was likely sufficient connectivity between Caribbean and Pacific oceans to maintain dispersal for some octopus species between these locations up until as recently as the Early Pliocene.

The emergence of the Isthmus of Panama was a long process that caused profound but gradual changes in a range of oceanographic conditions, including temperature, salinity, circulation and productivity (O’Dea et al., 2016). The seaway is understood to have been shallowing by the Middle Miocene, decreasing in depth from over 2,000 m to less than 1,000 m deep (Coates, 1997). Reorganisations of ocean circulation from eastward-flowing to westward-flowing occurred during the Late Oligocene increased productivity within the Caribbean during the Early Miocene (Bartoli et al., 2005; Schneider & Schmittner, 2006). Temperature and salinity began to increase approximately 7 Ma (Collins, 1996). Around 4 Ma, the narrowing of the seaway began to extinguish Caribbean upwelling and the primary productivity of this region dropped dramatically, while it increased in the Eastern Pacific (Pennington et al., 2006; Lessios, 2008; Coates & Stallard, 2013; Leigh, O’Dea & Vermeij, 2013).

Considering that the oceanographic changes were so strong as to affect the global climate (Lear, Rosenthal & Wright, 2003), we suggest that octopods TSSP/TSSC probably diverged as a consequence of these physical and environmental barriers, even before the complete uplift of the IP 3 Ma suggested by the Late Pliocene model. A revision carried out by Lessios (2008) indicated that 73 from 115 geminate clades, including echinoids, crustaceans, fishes and molluscs, split earlier than the final closure of the IP. A similar result was verified by O’Dea et al. (2016), who used 38 comparisons based on fossil-calibrated phylogenies and revealed that 26 (68%) produced estimates of separation that occurred more recently than 12 Ma. Marko (2002), studying divergences among six pairs of geminates in the Arcidae bivalves, also reported that isolation of geminate species did not necessarily occur in the latest stages of closure of the Central American Seaway.

The soft-bodied cephalopods are poorly represented in the fossil record, which makes estimating divergence times of octopuses challenging (Strugnell et al., 2006). However, divergence times of non-calibrated nodes generated in the present study are consistent with previous studies. Using two biogeographical calibrations (LGM and uplift of the IP), Gleadall (2013) estimated that Enteroctopus and Muusoctopus lineages separated 22 +/ − 2.2 Ma, which is in accordance with this study (22 Ma, 10-40 95% HPD in this study). The age of separation between the M. januari and M. yaquinae clades (11 Ma, 5–21 95% HPD) is also in accordance with Gleadall’s (2013) results (13.4 Ma). Furthermore, Amor et al. (2014), using the rate of evolution for COI in cephalopods, estimated 19.0–40.9 Ma for the segregation between the O. vulgaris and O. mimus groups, which is in accordance with the results of this study (29 Ma, 18–43 95% HPD).

According to the reconstruction ancestral area analysis, the most recent common ancestors of all octopod TSSP/TSSC originated from the Pacific Ocean. The shift in the ocean circulation flow from east–west to west–east would carry paralarvae of many species from the Pacific to the Atlantic before the final closure of the IP (Bartoli et al., 2005; Schneider & Schmittner, 2006). Several studies have proposed that many Caribbean species of bivalve, gastropod and fish also derived from Pacific ancestors (Bermingham, Shawn McCafferty & Martin, 1997; Meyer, 2003; Leigh, O’Dea & Vermeij, 2013; LaBella et al., 2017). Additionally, Leigh, O’Dea & Vermeij (2013) pointed out that fossil evidence indicates a Pacific origin for six gastropod species that occupied both sides of tropical America and have become extinct in the Atlantic.

The earliest divergence of all octopods TSSP/TSSC investigated in this study occurred between the pygmy octopuses P. digueti from Eastern Pacific and P. mercatoris/P. cf. joubini from the Western Atlantic 17.44 Ma (8.99, 30.31 95% HPD, clade 3). Lower fecundity rates (20–320 eggs and benthic hatchling) (Forsythe & Toll, 1991) combined with small adult size (body weight from 20 g to 85 g) (Norman, Hochberg & Finn, 2014) may have reduced the ability of these species to maintain dispersal after the first environmental changes resulting from the formation of the IP.

The divergence among the deep-sea species M. januari and M. yaquinae group was around 11 Ma (5.08, 20.77 95% HPD, clade 1). This split was probably influenced by the shoaling of the IP, which shut off deep water connection between 12 and 9.2 Ma (Coates et al., 1992; Lear, Rosenthal & Wright, 2003). The deep divergences among transisthmian species can also be explained by the extinction of geminates as a consequence of the uplift of the isthmus (Leigh, O’Dea & Vermeij, 2013; Lessios, 1998). Marko (2002) affirms that molluscan transisthmian taxa from the EP and the WA may be rare due to successive events of extinction caused by the seaway closure, in which at least 70% of some molluscan subgeneric groups were lost.

The clade that included Macrotritopus defilippi from the Western Atlantic and Octopodidae sp. (White V) (Norman, 2000) from the Western Pacific shared a MRCA up to 8 Ma (2.82, 15.88 95% HPD, clade 4). These sand-dwelling species have similar morphological and complex behavioural characteristics, such as mimicry and ‘flatfish swimming’ (octopus that mimicked the shape, swimming actions, speed, duration and the colouration of swimming flounders) (Hanlon, Conroy & Forsythe, 2008). They also appear to have evolved from a sand-dwelling common ancestor with extremely long arms (Huffard et al., 2010) and with small and planktonic eggs at the Pacific Ocean. However, these relationships and routes of dispersal may be biased because the M. defilippi from the Mediterranean Sea (locality of the species holotype) was not available for this analysis.

The most recent divergence in our analysis is the nocturnal Callistoctopus species from the Western Pacific and Western Atlantic at 5 Ma (2.21, 10.96 95% HPD, clade 2). Callistoctopus ornatus occupies a broad area of the Indian and Western and Central Pacific Oceans, while C. macropus occurs in the Mediterranean Sea and Eastern Atlantic Ocean (Norman, Hochberg & Finn, 2014; Voss, 1981). The nocturnal Callistoctopus sp. from the Brazilian coast is morphologically, genetically and behaviourally related to C. ornatus and C. macropus (Leite TS, Lima FD, pers. comm., 2018). Additionally, all the species in clade 2 have small and planktonic hatchlings, which may have facilitated their dispersal and diversification across the seas (Boyle & Rodhouse, 2005; Norman, Hochberg & Finn, 2014). The BBM analysis pointed to the WP–WA as more plausible dispersal route of the nocturnal species, suggesting that the channel before the final closure of the IP was the likely pathway. However, the posterior probability is low (PP BBM route = 0.37), which suggests that alternative routes of colonisation, such as from Indo-Pacific to Eastern Atlantic, were also likely to have been taken. Schneider & Schmittner (2006) pointed out a connection between the tropical gateways of Panama and Indonesia in the way that reduced outflow of upper Pacific Ocean waters via the Panama seaway into the Atlantic is compensated by increased flow towards the Indonesian Archipelago. Given that the divergence of these species occurred shortly before the total closure of the isthmus, this interflow may explain the dispersal of species from the West to East Pacific and, subsequently to the West Atlantic via the Panama Seaway.

Clade 5 includes Octopus species with remarkable evolutionary success in term of diversification, distribution and abundance on both sides of the Americas. This TSSC groups includes O. maya, O. hummelincki and O. insularis from the WA and O. mimus and O. hubbsorum from the EP, which have different reproductive strategies and shared a common ancestor 8 Ma (4.27, 13.58 95% HPD). The species O. mimus and O. hubbsorum are closely related and may represent a single species (Pliego-Cárdenas et al., 2014). They are the putative transisthmian sister taxa of O. insularis, because they also share similar morphological characteristics (medium/large muscular species, ocellus absent, white spots on dorsal mantle, skin texture of irregular patches and a groove system), habitat preferences (reefs and rocky bottoms in shallow waters) and life history (small and planktonic eggs) (Leite et al., 2008; Leite et al., 2009; Norman, Hochberg & Finn, 2014).

Prior to the divergence of the species in clade 5, speciation processes appear to have occurred within the Eastern Pacific around 19 Ma (10–29 95% HPD). These processes led to a clade containing two ocellated species with different reproductive traits (O. bimaculoides and O. bimaculatus, holobenthic and pelagobenthic hatchlings, respectively) in EP and another clade containing the endemic ocellated species from the Galapagos, O. oculifer (holobenthic mode), and the species of clade 5. The divergence between O. oculifer and clade 5 (12 Ma, 7–20 95% HPD) occurred slightly after the formation of seamounts in Galápagos at 14.5 Ma (Werner et al., 1999), suggesting that this species successfully settled around the rising island and became endemic due to its low dispersive ability.

Even though the divergence of O. oculifer was not precipitated by a vicariance event related to the formation of IP, the subsequent evolutionary processes were and seem to have generated a species with very similar characteristics in WA, O. maya. They share important biological features, such as large eggs and benthic hatchlings (Arreguín-Sánchez, Solís & González De La Rosa, 2000), which means they probably inherited these traits from a common ancestor. Although some studies have pointed O. bimaculatus as the transisthmian sister taxon of O. maya (Voight, 1988; Juárez, Rosas & Arenta-Ortiz, 2012; Nesis, 2003; Allcock, Lindgren & Strugnell, 2015), this study suggests that O. oculifer is the sister transisthmian species of O. maya, since they also share morphological, behavioural and ecological similarities.

Conclusions

The long geological history of the Isthmus of Panama had an immense impact on the speciation processes of marine biota in the Atlantic and Pacific Oceans (Coates et al., 1992; Bartoli et al., 2005; O’Dea et al., 2007; Coates & Stallard, 2013; O’Dea et al., 2016). The divergence processes among octopods TSSP/TSSC probably occurred up to 5 Ma (2.21 Ma, lower bound of the 95% HPD interval), a long time after the final closure of the IP proposed by the Miocene model, 15 Ma. Considering the influence of the extreme environmental changes during this geological event in the speciation processes, this study indicates that the divergence times of the octopod species are according to the classic Pliocene model.

The results obtained in this phylogenetic reconstruction suggest that it is more probable that the differences among the lineages of octopod transisthmian species arose due to the allopatric speciation process caused by the uplift of Isthmus of Panama than that they arose due to an independent process of evolutionary convergence.

Supplemental Information

Figure S1 Bayesian phylogenetic tree including all cephalopods species used in this study

The bars on the nodes represent the 95% Highest Posterior Density intervals. The 95% HPD of calibrated nodes with three fossils information are shown in orange bars. The asterisk represents the biogeographical calibration. The mean ages of clades divergence are placed below each node (Ma).

Click here for additional data file.

Table S1 Details of the specimens for COI, 16S rDNA, Rhodopsin and EF1-alpha genes used to construct the final Bayesian phylogenetic tree in this study

GB = GenBank accession number, MORG = Museu Oceanográfico do Rio Grande, CTR = Coleção de tecidos de invertebrados da UFRN.

Click here for additional data file.

Supplemental Information 3 Sequences of 16S mitochondrial genes

Click here for additional data file.

Supplemental Information 4 Sequences of COI mitochondrial genes

Click here for additional data file.

Supplemental Information 5 Sequences of EF-1 alpha nuclear genes

Click here for additional data file.

Supplemental Information 6 Sequences of rhodopsin nuclear genes

Click here for additional data file.

We are thankful to the Chico Mendes Institute for Biodiversity Conservation (MMA/ICMBIO) and the Brazilian Navy for logistics support in the field work. We are also grateful to Carlos Rosas, Jaciana Barbosa, Diego Batista, Leocledna Oliveira and Juliana Valverde for helping in the data collection.

Additional Information and Declarations

Competing Interests

Author Contributions

Animal Ethics

Field Study Permissions

Data Availability

The authors declare there are no competing interests.

Francoise D. Lima conceived and designed the experiments, performed the experiments, analyzed the data, prepared figures and/or tables, authored or reviewed drafts of the paper, and approved the final draft.

Jan M. Strugnell and Sergio M.Q. Lima conceived and designed the experiments, analyzed the data, authored or reviewed drafts of the paper, and approved the final draft.

Tatiana S. Leite conceived and designed the experiments, authored or reviewed drafts of the paper, taxonomy details, and approved the final draft.

The following information was supplied relating to ethical approvals (i.e., approving body and any reference numbers):

The data collected in this study were obtained from fish markets in Brazil and Mexico. Most of the animals collected in this study were obtained from fish markets and samples from universities (James Cook University).

The following information was supplied relating to field study approvals (i.e., approving body and any reference numbers):

Field experiments were approved by the Instituto Chico Mendes de Conservação da Biodiversidade, ICMBIO (107065 and 304841).

The following information was supplied regarding data availability:

The sequences are available in GenBank (accessions are listed in Table S1).

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
