# Peer review of "A biogeographic framework of octopod species diversification: the role of the Isthmus of Panama"

_PeerJ, doi:10.7717/peerj.8691_

## Round 0.1 · original submission · Major Revisions

As you will see, the manuscript has been read and commented upon by two expert reviewers. Both had favorable things to say and the reviewers’ points will be relatively easily addressed. Nevertheless, I have ranked this as a “major revision” because two items raised by Reviewer 2 are of vital importance and need to be addressed before this can be accepted: (1) the inclusion of Genbank accession numbers (which was probably already planned) and (2) linking this study to numbered voucher specimens in accessible museum collections. Please respond to all the individual comments by the reviewers and make sure that the final manuscript has been reviewed by a native English speaker before resubmission. I look forward to receiving the new version!

Reviewer 1 ·

Basic reporting

The authors use a phylogenetic approach to test the influence of the closure of the Isthmus of Panama on speciation processes in octopods. The paper is excellent, I really enjoyed reading it. It will represent a very nice addition to the existing litterature on the topic. Working hypothese are clearly stated, the experiment was well conducted with a large sample size across a broad geographical area and the analytical approach is sound and robust.
The authors placed their findings in the context of existing litterature and provided sufficient background. Figures/tables are clear, and the raw data is available. My only suggestion would be to have the text reviewed by a native engish speaker to correct small language mistakes throughout.

Experimental design

This work is very relevant, and performed to a high technical standard. Questions and aims are well defined and methods are clearly explained,

Validity of the findings

Results are robust and statistically sound. Conclusions are well stated and results are discused in the context of existing litterature.

Reviewer 2 ·

Basic reporting

The article is clear and the analyses are presented in a straight forward manner.

Genbank accession numbers are not provided in Table 1 as stated. The accession number range should be included in the main text, and the breakdown of accession numbers inserted into Table S1.
Equally importantly, no voucher specimens are referred to. Given the difficulties of correct identification of octopod species, it is essential that vouchers are submitted to museums for all the newly sequenced material. I suggest adding an extra column to S1 to accommodate this.

Please explain why the last occurrence of ammonites controls the upper bound of argonauts given these two groups are very unrelated. Generally, more detail in the fossil constraints would be helpful. For example, why was Vampyronassa chosen? No reason is given.

Figure 1 contains several errors.
Authors are advised to follow the convention of using two letters for genus abbreviations where they include more than one genus starting with the same letter. For example, use Op. for Opisthoteuthis, and Oc. for Octopus. There is a general inconsistency in the figure over using genus abbreviations (or not). "longibrachus" is spelt incorrectly in the eastern Atlantic.

Authors should take care to cite early works and credit those authors that had the earliest thoughts about Trans-Panamanian geminate species, not just recent papers.

Attention to English language is required throughout. Examples of errors included "waters exchanges", "octopods species", "stripped" instead of "striped", "comprise of" rather than the correct "comprises", misplaced definite articles, and incorrect pronouns. Sentences on lines 118-120, 393-397, and 406-409 don't make sense. Some family names are italicised, and some genera are not.

Experimental design

Authors are commended for including three of the four species pairs recommended as potentially trans-Panamanian by Naef, 1978.

Validity of the findings

The 95% HPD limits are very wide, with the highest bound of each clade ranging between 11 and 30 Ma, thus authors should take care how they interpret the data. Line 401 they state that divergence processes occurred until 5 Mya but that depends rather on where the true date lies in their 95% HPD.

Additional comments

The second author published a paper revising octopod taxonomy, yet the authors do not adhere to that new (and as far as I am aware accepted) systematic scheme. Octopodidae now refers to a restricted group and does not include Muusoctopus species. Authors are recommended to carefully revise the text so that their higher taxonomy reflects WoRMS and Strugnell et al. 2014.

"dispersion" has a very special meaning in biology related to the spatial distribution (uniform, random, clumped). It should not be used as a synonym for dispersal.

Line 357, no species have planktonic eggs. Octopuses lay their eggs attached to substrate and these may hatch into planktonic paralarvae in some species.

Please ensure that the way abbreviations are used in a sentence makes sense when they are expanded and that abbreviations are correctly pluralised. Failure to do this makes the text difficult to follow.

---

## Round 0.2 · Minor Revisions

Your manuscript has been reviewed again by one of the original reviewers who repeats two points that should be addressed (incomplete GenBank references and number of species in the family Octopodidae) with which I agree. Please also note the disparity between text and table as pointed out by the reviewer.

I noticed that your additions have introduced new ambiguities, probably due to language issues: I am not entirely sure what you mean by “This species probably inherited these traits from the same ancestor that shared with O. oculifer in the past” (lines 414-415) and by “shared a recent common ancestral [should this be ancestor?] from the Pacific Ocean” (line 429). You also newly introduced odd phrases such as “verify the whether the divergence time” (lines 125-126) and “fossil records of argonaut" (line 159).

Reviewer 2 ·

Basic reporting

While generally very well written and clear, there remain many places where language correction is required. The paper needs to be proof read and corrected by the native English speaker among the authors.

There is a disparity between the text and table 2 (17.24 or 17.44)

Experimental design

The science is sound

Validity of the findings

No further comments to make

Additional comments

The science in this appears to be sound and it will be a useful addition to the literature. There are just a couple of points raised in the previous review that have not been addressed:

My previous comment: "Genbank accession numbers are not provided in Table 1 as stated. The accession number range should be included in the main text, and the breakdown of accession numbers inserted into Table S1."

The authors have included the Genbank accessions they have in Table S1 and will add others in proof. But Supplementary material becomes separated from the paper over time (as academics share pdfs) and it is essential that the range of Genbank numbers (e.g., COI FZ654009-FZ654092; 16S FZ654093-FZ654142 etc) be included in the main text. This does not take up much space but will be enormously useful to future readers.

My previous comment: "The second author published a paper revising octopod taxonomy, yet the authors do not adhere to that new (and as far as I am aware accepted) systematic scheme. Octopodidae now refers to a restricted group and does not include Muusoctopus species."

The authors say “We described the Family of Muusoctopus species in the line 249”, but this doesn’t address my point. In line 106, the authors say “The greatest diversity of benthic octopuses occurs in the Family Octopodidae with more than 300 species world-wide”. The authors here, following the taxonomic reorganisation of Strugnell et al. 2014, are referring to the Superfamily Octopodoidea (this is clear because if they are referring to Octopodidae, i.e., excluding Enteroctopodidae, Eledonidae, Megaleledonidae etc, then there aren’t more than 300 species - more like 150…)

---

## Round 0.3 · accepted · Accept

Thank you for addressing the queried issues in the revised version. I am happy to accept your manuscript.